# Redox Regulation of Microglial Inflammatory Response: Fine Control of NLRP3 Inflammasome through Nrf2 and NOX4

**DOI:** 10.3390/antiox12091729

**Published:** 2023-09-07

**Authors:** Alejandra Palomino-Antolín, Céline Decouty-Pérez, Víctor Farré-Alins, Paloma Narros-Fernández, Ana Belen Lopez-Rodriguez, María Álvarez-Rubal, Inés Valencia, Francisco López-Muñoz, Eva Ramos, Antonio Cuadrado, Ana I. Casas, Alejandro Romero, Javier Egea

**Affiliations:** 1Unidad de Investigación, Hospital Santa Cristina, Instituto de Investigación Sanitaria Princesa (IIS-IP), 28006 Madrid, Spain; alejandra.palominoantolin@gmail.com (A.P.-A.); celine.decouty@estudiante.uam.es (C.D.-P.); victorfarre@hotmail.com (V.F.-A.); narrosfp@tcd.ie (P.N.-F.); anabelen.lopez.externo@salud.madrid.org (A.B.L.-R.);; 2Faculty of Health, Camilo José Cela University of Madrid (UCJC), 28692 Madrid, Spain; 3Neuropsychopharmacology Unit, Hospital 12 de Octubre Research Institute, 28041 Madrid, Spain; 4Department of Pharmacology and Toxicology, Faculty of Veterinary Medicine, Complutense University of Madrid, 28040 Madrid, Spain; eva.ramos@ucm.es; 5Instituto de Investigaciones Biomédicas “Alberto Sols” UAM-CSIC, Centro de Investigación Biomédica en Red Sobre Enfermedades Neurodegenerativas (CIBERNED), Instituto de Salud Carlos III (ISCIII), 28031 Madrid, Spain; antonio.cuadrado@uam.es; 6Pharmacology and Personalised Medicine, Maastricht University, 6211 LK Maastricht, The Netherlands; 7Neurology Clinic, University Hospital Essen, 45147 Essen, Germany; 8Center for Translational Neuro- and Behavioral Sciences (C-TNBS), 45147 Essen, Germany; 9Department of Neurology, University Hospital Essen, 45147 Essen, Germany

**Keywords:** inflammation and immunity, NLRP3 inflammasome, Nrf2, NOX4, glial cultures, KO mice, neurodegenerative diseases

## Abstract

The role of inflammation and immunity in the pathomechanism of neurodegenerative diseases has become increasingly relevant within the past few years. In this context, the NOD-like receptor protein 3 (NLRP3) inflammasome plays a crucial role in the activation of inflammatory responses by promoting the maturation and secretion of pro-inflammatory cytokines such as interleukin-1β and interleukin-18. We hypothesized that the interplay between nuclear factor erythroid 2-related factor 2 (Nrf2) and NADPH oxidase 4 (NOX4) may play a critical role in the activation of the NLRP3 inflammasome and subsequent inflammatory responses. After priming mixed glial cultures with lipopolysaccharide (LPS), cells were stimulated with ATP, showing a significant reduction of IL1-β release in NOX4 and Nrf2 KO mice. Importantly, NOX4 inhibition using GKT136901 also reduced IL-1β release, as in NOX4 KO mixed glial cultures. Moreover, we measured NOX4 and NLRP3 expression in wild-type mixed glial cultures following LPS treatment, observing that both increased after TLR4 activation, while 24 h treatment with tert-butylhydroquinone, a potent Nrf2 inducer, significantly reduced NLRP3 expression. LPS administration resulted in significant cognitive impairment compared to the control group. Indeed, LPS also modified the expression of NLRP3 and NOX4 in mouse hippocampus. However, mice treated with GKT136901 after LPS impairment showed a significantly improved discrimination index and recovered the expression of inflammatory genes to normal levels compared with wild-type animals. Hence, we here validate NOX4 as a key player in NLRP3 inflammasome activation, suggesting NOX4 pharmacological inhibition as a potent therapeutic approach in neurodegenerative diseases.

## 1. Introduction

Inflammation is a protective response to infection or injury characterized by the recruitment and activation of immune cells to fight infection and remove debris. The inflammatory response eliminates harmful stimuli and restores tissue homeostasis through tissue and wound repair. Excessive inflammation contributes to chronic inflammatory disease, which can also lead to sepsis and subsequent multiorgan failure [1]. NOD-like receptor protein 3 (NLRP3) inflammasome is a cytoplasmic multiprotein complex that plays a crucial role in regulating the innate immune response [2]. Activation of the NLRP3 inflammasome requires two steps: priming and activation. The priming step involves the transcriptional upregulation of NLRP3 and pro-interleukin-1β (IL-1β) genes by nuclear factor-kappa B (NF-κB) in response to pathogen-associated molecular patterns (PAMPs) or danger-associated molecular patterns (DAMPs) [3]. The second step involves the assembly and activation of NLRP3 inflammasome complex, which activates caspase-1. This caspase-1 activation leads to the maturation and secretion of proinflammatory cytokines, such as IL-1β and interleukin-18 (IL-18). Additionally, GSDM is also cleaved by caspase-1, forming pores in the membrane and, consequently, pyroptosis [4,5]. Dysregulation of NLRP3 inflammasome activation has been implicated in the pathogenesis of various diseases, including neurodegenerative [6], diabetes, and autoimmune diseases [7].

Reactive oxygen species (ROS) are highly reactive molecules that are produced during normal cellular metabolism. They play an essential role in cellular signaling and regulation of various physiological processes such as cell proliferation, differentiation, cellular senescence, and apoptosis [8]. However, the excessive production of ROS can cause oxidative stress, which can lead to cellular damage, aging, and various diseases [9]. Therefore, it is crucial to maintain a balance between ROS production and scavenging to prevent the deleterious effects of oxidative stress. The production and scavenging of ROS are tightly regulated by a complex network of enzymes and transcription factors. 

NADPH oxidases (NOX) are considered an important enzymatic sources of ROS production [10]. Of these, NOX4 appears to be the most suitable therapeutic target because it is induced in various cells and tissues under ischemia or hypoxia [11]. It catalyzes the transfer of electrons from NADPH to molecular oxygen to produce a superoxide anion (O_2_^•^) and hydrogen peroxide (H_2_O_2_), which are the primary ROS produced by NOX4. In this regard, NOX4-derived ROS have been shown to play a critical role in the priming and activation of the NLRP3 inflammasome [12] by activating NF-κB signaling, leading to the upregulation of NLRP3 and pro-IL-1β genes [12]. Moreover, NOX4-derived ROS have been shown to activate the NLRP3 inflammasome directly by inducing potassium efflux and calcium influx, leading to the activation of caspase-1 and the subsequent release of IL-1β and IL-18. On the other hand, nuclear factor erythroid 2-related factor 2 (Nrf2) is a transcription factor that is found imprisoned in the cytoplasm, but under stress conditions such as the formation of ROS, its translocation to the nucleus is induced, leading to the transcription of a battery of genes encoding antioxidant and detoxification enzymes [13]. In this context, Nrf2 plays a critical role in protecting cells from oxidative stress by upregulating the expression of various antioxidant genes, including glutathione peroxidase, superoxide dismutase, and catalase [13]. Additionally, Nrf2 has been shown to be a negative regulator of NLRP3 inflammasome activation by reducing ROS production and modulating the activity of NF-κB [14]. Therefore, the interplay between NOX4 and Nrf2 in the regulation of NLRP3 inflammasome ROS homeostasis is a complex and dynamic process. Moreover, NOX4-derived ROS can contribute to NLRP3 inflammasome activation, while Nrf2 activation can modify ROS levels and mitigate NLRP3 inflammasome activation. 

In this manuscript, we aimed to contribute to the understanding of the role of NOX4 and Nrf2 in the activation of the NLRP3 inflammasome. Our data show that (i) both NOX4 and Nrf2 participate in NLRP3 inflammasome activation; (ii) in NOX4 and Nrf2 KO animals, there was a reduction in the release of IL-1β in response to LPS plus ATP; (iii) NOX4 inhibition using GKT136901 also reduced IL-1β release as well as NOX4 KO mixed glial cultures. In addition, we measured NOX4 and NLRP3 expression in WT mixed glial cultures following LPS treatment, observing that both increased after TLR4 activation, while 24 h treatment with tert-butylhydroquinone (tBHQ), a potent Nrf2 inducer, significantly reduced NLRP3 expression. In vivo LPS administration resulted in significant cognitive impairment compared to the control group. However, mice treated with GKT136901 after LPS impairment showed a significantly improved discrimination index and recovery of the expression of inflammatory genes to normal levels. This shows that NOX4 is a key player in NLRP3 inflammasome activation. 

## 2. Materials and Methods

### 2.1. Mixed Glial Cultures

Mixed glial cultures were prepared from the cerebral cortices of postnatal day 3 C57BL/6N wild-type NOX4 knockout and Nrf2 knockout mice, as previously described [15,16]. Briefly, after removing the meninges and blood vessels, the forebrain was carefully dissociated in DMEM/F12 medium by repeated pipetting. After mechanical dissociation, cells were seeded in DMEM/F12 containing 20% FBS at a density of 300,000 cells/mL and incubated at 37 °C in humidified 5% CO_2_ and 95% air. After 5 days in vitro (DIV), the medium was replaced with DMEM/F12 and 10% FBS. Cultures were used at confluency reached after 10–12 DIV. Cultures were treated with LPS (1 μg/mL), ATP (5 mM) (Sigma-Aldrich, Madrid, Spain), tBHQ (10 μM), and GKT136901 (1 μM). 

### 2.2. Determination of IL-1β Levels in the Culture Medium

After the different drug treatments, IL-1β levels were measured by using a specific ELISA kit. Supernatant samples were collected at the indicated time points and subjected to the ELISA analysis according to the supplier’s recommendations (DY401, R&D Systems, Minneapolis, MN, USA).

### 2.3. Animals

All animal experimentation was performed under the license PROEX 013/18 granted by the Ethics Committee of Universidad Autónoma de Madrid (Madrid, Spain) and in compliance with the Cruelty to Animals Act, 1876, and the European Community Directive, 86/609/EEC. Every effort was made to minimize stress to the animals. Animals were housed under controlled conditions (22 ± 1 °C, 55–65% humidity, 12 h light-dark cycle), and have been given free access to water and standard laboratory chow. Experiments were performed on 3–4-month male C57BL/6N wild-type mice, NOX4 KO mice, C57BL/6J, and Nrf2 KO mice. NOX4 KO mice were generated in 2010 by deleting the NADPH binding pockets located in exons 14 and 15 to directly assess NOX4 function in vivo without modifying its expression [11]. In this work, Kleinschnitz C. et al. verified that NOX4 in these animals had no activity. For our experiment, animals were randomly divided into the different experimental groups (vehicle, LPS 250 µg/kg, LPS 250 µg/kg + GKT136901 10 mg/kg). A mixture of DMSO/water in a ratio of 1/99 (10 mL/kg) was used as a vehicle. All the treatments were injected intraperitoneally (i.p.). For the analyses of protein and transcriptional changes, animals were terminally anesthetized with a mix of ketamin:xylacin 1:2 (Ketolar 50 mg/mL, Pfizer, Madrid, Spain; Xilagesic 20 mg/mL, Calier Labs, Barcelona, Spain) and transcardially perfused with heparinized saline. Hippocampi were gently removed and stored at −80 °C until use.

### 2.4. Novel Object Recognition Test in Mice

The novel object recognition (NOR) test is a behavioral test commonly used to assess recognition memory in mice [17]. For 3 consecutive days, the animals were placed on a field (40 × 40 × 40 cm, made of polyvinyl chloride) for 10 min. On the first day (T0), mice explored an empty box. On day 2 (T1), the animal was placed in the field with two identical objects (cylindrical glass bottles, too heavy for the mouse to move, 22 cm high, 9 cm in diameter) and allowed to explore for 10 min. On the third day (T2), a new object (new object, square object) was placed in place of one of the old objects (familiar object) and the other was left as is. Object exploration was measured using a stopwatch with the animal sniffing, wiping, or looking at the object from a distance of up to 2 cm. All object positions were balanced between groups, and objects and NOR fields were washed with 0.1% acetic acid between trials to balance olfactory cues. The amount of time spent examining new or familiar objects was videotaped for 10 min and scored by a blind observer. NOR is based on the premise that rodents have an innate preference for new objects, so mice that remember familiar objects spend more time exploring new objects. Discrimination index in the T2 was estimated as follows: Discrimination Index (%) = [Time exploring novel object − Time exploring familiar object)/(Time exploring novel object + Time exploring familiar object)] × 100. LPS (250 μg/kg) was injected intraperitoneally (i.p.) immediately after the end of the T1 phase. GKT136901 was dissolved in a mixture of DMSO/water in a ratio of 1/99. GKT136901 5 (10 mg/kg i.p.) was administered immediately after the end of the T0 and T1 phases.

### 2.5. Quantitative Real-Time PCR

Total RNA was extracted from mixed glial cultures and hippocampi using the TRIzol method (10296-028, Invitrogen, Carlsbad, CA, USA). cDNA was synthesized using the iS-cript cDNA synthesis kit (1708891, Biorad, Hercules, CA, USA) according to the manufacturer’s instructions. Quantitative polymerase chain reaction (qPCR) was performed using the 7900HT Fast Real-Time PCR System (Applied Biosystems, Waltham, MA, USA) in 384-well format with Power SYBR Green PCR Master Mix (Thermo Fisher, Waltham, MA, USA). Data were normalized to the expression of the housekeeping gene B2M (NM_009735). Specific primers were designed using the NCBI nucleotide data base and Primer 3 software (Version 4.1.0) (http://biotools.umassmed.edu/bioapps/primer3_www.cgi (accessed on 3 February 2023). Primer sequences were checked with BLAST (http://blast.ncbi.nlm.nih.gov/Blast.cgi (accessed on 3 February 2023)) and purchased from Sigma or IDT. The comparative CT method (or the 2^ΔCT^ method) [18,19] was used to determine differences in gene expression between B2M and control samples. The target genes and the specific primers were the following: Nlrp3 (NM_145827) (forward, 5′-TTCAATCTGTTGTTCAGCTC-3′; reverse, 5′-GTCTAATTCCAGCCATCTGTAG-3′), NOX4 (NM_001285835) (forward, 5′-GGGACATTAAACGATTAAACAAGAATCC-3′); reverse, 5′-GGAAGTATTGGCTTCTTATTGG-3′), Il1b (NM_008361) (forward, 5′-GAAGAGCCCATCCTCTGTGA-3′; reverse, 5′-TTCATCTCGGAGCCTGTAG-3′). 

### 2.6. Immunoblotting and Image Analysis

After the different treatments, mixed glial cultures and mouse hippocampi were lysed in ice-cold lysis buffer (1% Nonidet P-40, 10% glycerol, 137 mM NaCl, 20 mM Tris–HCl, pH 7.5, 1 g/mL leupeptin, 1 mM PMSF, 20 mM NaF, 1 mM sodium pyrophosphate, and 1 mM Na_3_VO_4_). Proteins (30 µg) from cell lysates were separated via SDS-PAGE and transferred to Immobilon-P membranes (Millipore Corp., Billerica, MA, USA). Membranes were incubated with anti-NLRP3 (1:1000; AdipoGen, San Diego, CA, USA), anti-NOX4 (1:1000; R&D Systems, Minneapolis, MN, USA), anti-NQO1 (1:1000; Cell Signaling Technology, Danvers, MA, USA) or anti-βactin (1:50,000; Sigma-Aldrich, Madrid, Spain). Appropriate peroxidase-conjugated secondary antibodies (1:5000; Santa Cruz Biotechnology, Dallas, TX, USA) were used for protein detection via enhanced chemiluminescence on the ImageQuant LAS 4000 min. Different band intensities corresponding to immunoblot detection of protein samples were quantified using the Scion Image program. Immunoblots corresponded to a representative experiment that was repeated three or four times with similar results.

### 2.7. Immunocytochemistry 

For the immunofluorescence experiments, primary glial cells were always seeded in 24-well plates, and each well required a coverslip (Ø 13 mm; VWR, Leicestershire, UK), in which the cells grew. After the different treatments were applied, it was necessary to wash three times with PBS 1X and then fix the cells with 4% paraformaldehyde in PBS 1X. The next step was to permeabilize the cells with PBS 0.3% Triton for 10 min at room temperature, then apply a blocking solution containing PBS 1X plus 10% bovine serum albumin (BSA), for 1 h. Primary antibody was diluted in PBS 1X with 1% BSA and incubated overnight at 4 °C: NRF2 (1:100; ref). The next day, we applied the secondary antibody donkey anti-rabbit Alexa Fluor 488 (1:800, ThermoFisher Cat# A21206) for 1 h at room temperature. After incubation in the secondary antibody, the sections were washed four times for 15 min each in PBS, mounted with a small drop of VECTASHIELD mounting medium containing DAPI (Vector Laboratories Inc., Newark, CA, USA, Cat. No. H-1000), and the slides were cover-slipped. All immunostained brain images were acquired with a Leica SP5 confocal microscope. Images were processed with the program ImageJ 1.52e.

### 2.8. Statistical Analysis

For multiple comparisons, one- or two-way analysis of variance (ANOVA) was used, with the factors being genotype (wild type (WT) or NOX4 or Nrf2 knockout (KO)) and treatment (vehicle or GKT136901 treatment). Subsequent post hoc comparisons (Bonferroni’s test) were performed with a level of significance set at *p* < 0.05. Data are presented as mean ± standard error of the mean (SEM). Symbols in the graphs denote post hoc tests. Statistical analyses were carried out with the SPSS 22.0 software package (SPSS, Inc., Chicago, IL, USA).

## 3. Results

### 3.1. Genetic and Pharmacological Deficiency of NOX4 Reduces NLRP3 Inflammasome Components and IL-1β Release in Primary Mixed Glial Cultures 

In order to investigate the relationship between NLRP3 inflammasome and NOX4, we used a well-characterized model of NLRP3 inflammasome activation (LPS 4 h + ATP 30 min) in WT and in NOX4 deficient mice, and we also used the pharmacological inhibitor of NOX4 (GKT136901). We measured IL-1β release in mixed glial cultures using the protocol shown in Figure 1A. We used LPS for 4 h to increase the expression of NLRP3 inflammasome components, a well described process called “priming”. In the last 30 min, we added ATP to allow NLRP3 oligomerization and activation, and, finally, the release of IL-1β to the culture medium. Stimulation of WT cultures with LPS (1 µg/mL, 4 h) plus ATP (5 mM, 30 min) produced a significant release of IL-1β (Figure 1A). However, in NOX4 KO cultures we observed a significant reduction of IL-1β release compared to WT (60% reduction). Moreover, in wild-type cultures treated with the same stimuli, we observed the same effect using GKT136901 (1 µM), a selective pharmacological inhibitor of NOX4 (Figure 1A). To better explore the mechanism of NLRP3 inflammasome inhibition, we measured NLRP3 inflammasome components (Nlrp3 and Il1b) with RT-PCR using the same protocol. As illustrated in Figure 1B, LPS produced the increase in Nlrp3 and Il1b mRNA both in WT and NOX4 KO mice, and GKT136901 significantly reduced Nlrp3 and Il1b mRNA levels only in WT mice. 

Next, we checked the levels of NLRP3 using WB to confirm the data obtained by RT-PCR. Figure 2 shows the Western blot of NLRP3 in WT and NOX4 KO mice in the different conditions. Incubation of cells with LPS for 2 and 4 h significantly increased the production of NLRP3. Co-treatment with GKT136901 partially reduced NLRP3 protein levels in WT mice but not in NOX4 KO mice, confirming the results obtained in mRNA. Together, these data indicate that NOX4 inhibition is important for NLRP3 inflammasome activation in mixed glia cultures.

### 3.2. NOX4 Activity Resulted in Nrf2 Translocation to the Nucleus That Is Necessary for NLRP3 Inflammasome Activation

As stated in the Introduction, NOX are one of the main sources of ROS and, importantly, the only known enzyme family that has ROS formation as its sole known function [20]. Of these, NOX4 appears to be the most promising target for various diseases like neurodegenerative diseases or brain ischemia [21]. On the other hand, Nrf2 is known as a master regulator of antioxidant and anti-inflammatory responses, and in the presence of oxidative stress, Nrf2 translocates into the nucleus to induce phase II antioxidant response, a set of key proteins that detoxify xenobiotics. Hence, we wanted to explore whether NOX4-derived ROS produced by LPS treatment can induce the translocation of Nrf2 to the nucleus. To confirm this hypothesis, we evaluated the nuclear translocation of Nrf2 induced by 4 h treatment with LPS. Mixed glial cultures from WT and NOX4 KO mice were treated with LPS for 4 h, then were fixed and double-stained with anti-Nrf2 and DAPI. As shown in Figure 3A,B, in control conditions, Nrf2 was predominantly present in the cytosol; however, in the presence of LPS 4 h Nrf2 was predominantly located in the nucleus in WT cultures (Figure 3A) but not in NOX4 KO animals (Figure 3B). We used tBHQ as positive control of Nrf2 translocation. tBHQ treatment resulted in the translocation of Nrf2 to the nucleus both in WT and in NOX4 KO cultures. To corroborate the induction of phase II enzymes by LPS treatment, we analyzed the protein levels of NAD(P)H quinone oxidoreductase 1 (NQO1) (Figure 3C), one of the phase II enzymes induced by Nrf2. LPS treatment significantly increased NQO protein levels by 2-fold and co-incubation with GKT136901 partially reduced the increase in NQO1. Together these results suggest that Nrf2 participates in NOX4-derived ROS activation of NLRP3 inflammasome. 

### 3.3. Nrf2 Is Required for NLRP3 Inflammasome Activation

Next, we used the protocol shown on top of Figure 4 to determine the participation of Nrf2 in NLRP3 inflammasome activation. Stimulation of WT cultures with LPS (1 µg/mL, 4 h) plus ATP (5 mM, 30 min) produced a significant release of IL-1β (Figure 4A). However, in Nrf2 KO cultures, we did not observe any IL-1β release. Furthermore, we observed the same blocking effect on IL-1β release using a 24 h pre-treatment with tBHQ (10 µM) after LPS plus ATP (Figure 4A). To explore further in the mechanism of NLRP3 inflammasome, we measured NLRP3 inflammasome components (Nlrp3 and Il1b) by RT-PCR using the same protocol. As illustrated in Figure 4B, LPS resulted in an increase in Nlrp3 and Il1b mRNA both in WT and NOX4 KO mice, and 24 h pre-treatment with tBHQ significantly reduced Nlrp3 and Il1b mRNA levels in both WT and NOX4 KO mice. Together, these data indicate that Nrf2 is necessary for NLRP3 inflammasome activation in mixed glia cultures. 

### 3.4. NOX4 Genetic Deletion and Inhibition with GKT136901 Results in Memory Impairment Induced by LPS Administration In Vivo

Neurodegenerative diseases, such as Alzheimer’s and Parkinson’s disease, are characterized by several pathological features, including aggregation of specific proteins, selective neuronal loss, and chronic inflammation. To evaluate how NOX4 could be controlling neuroinflammation in vivo, we used the LPS-model that drives a transient sickness behavior response characterized by weight loss and memory loss without affecting locomotor activity [22]. We evaluated the memory loss induced by LPS (250 µg/kg) using WT and NOX4 KO mice and pharmacological inhibition of NOX4 with GKT136901 using the novel object recognition test (NOR) in mice, following the protocol shown in Figure 5A. In these conditions, administration of LPS significantly impaired NOR performance in WT animals (Figure 5B). Memory impairment produced by LPS injection was partially reversed with pharmacological treatment using the inhibitor of NOX4 GKT136901. Furthermore, in NOX4 KO animals, injection of LPS did not produce any memory impairment compared with NOX4 KO vehicle mice (Figure 5B). Taken together, the absence of NOX4 and its pharmacological inhibition improves memory impairment induced by LPS challenge in mice. 

### 3.5. NOX4 Genetic and Pharmacological Inhibition Modulate Inflammasome Component Expression In Vivo

The LPS model of transient inflammation is characterized by an acute inflammatory response during the first few hours and a high increase in proinflammatory cytokine expression, which remains elevated for up to 24 h. Therefore, following the experimental protocol of Figure 5, we evaluated NLRP3 inflammasome components (Nlrp3 and Il-1b) with RT-PCR in mouse hippocampus at 24 h post-LPS administration. At 24 h post-LPS injection, there was a significant increase in Nlrp3 and Il-1b mRNA levels in WT animals (Figure 6A,B). GKT136901 co-administration with LPS significantly reduced the expression of both Nrlp3 and Il-1b genes. Additionally, there was a significant increase in the mRNA levels of NOX4 in LPS-treated mice compared to vehicle in WT mice but not in NOX4 KO mice. Co-administration of GKT136901 reversed NOX4 mRNA increase near to vehicle levels (Figure 6C).

Finally, we wanted to confirm these results via Western blot in the hippocampus of animals 24 h post-LPS administration. At the protein level, we observed that there was an increase in NLRP3 and NOX4 amounts in LPS-treated WT animals (Figure 7). These increases were not observed in LPS-treated NOX4 KO animals or in WT animals treated with GKT136901, corroborating the results observed in mRNA. These results confirm that NOX4 is a key protein for NLRP3 inflammasome component expression and activation, and that its pharmacological inhibition can be a good pharmacological treatment to reduce inflammatory levels and improve memory impairment.

## 4. Discussion

We here validate NOX4 as a key player in NLRP3 inflammasome activation, suggesting NOX4 pharmacological inhibition as a potent therapeutic approach in neurodegenerative diseases. In this research, we contribute to the understanding of the role of NOX4 and Nrf2 in the activation of the NLRP3 inflammasome. Our data show that both NOX4 and Nrf2 participate in NLRP3 inflammasome activation, since (i) we showed a reduction in IL-1β release in response to LPS plus ATP in NOX4 and Nrf2 KO animals; (ii) pharmacological inhibition of NOX4 using GKT136901 and Nrf2 activation using tBHQ also reduced IL-1β release; (iii) we measured NOX4 and NLRP3 expression in WT mixed glial cultures following LPS treatment, observing that both increased after TLR4 activation, while 24 h treatment with tert-butylhydroquinone, a potent Nrf2 inducer, significantly reduced NLRP3 expression. In vivo LPS administration resulted in significant cognitive impairment compared to the control group. However, mice treated with GKT136901 after LPS impairment showed a significantly improved discrimination index in the NOR test and recovery of the expression of inflammatory genes to normal levels. Nevertheless, we should consider that our study has some limitations. In this study, to directly assess NOX4 function in vivo without altering its expression, we selected NOX4 KO mice generated in 2010 by deleting the NADPH binding pockets located in exons 14 and 15. In in vitro experiments, other NOX isoforms are present that may be involved in LPS-dependent ROS generation. Therefore, it would be of interest in future experiments to use selective inhibitors of the remaining isoforms to determine their involvement.

Aging is characterized by a gradual, cumulative deterioration in physiological functions over time which results in increased susceptibility to diseases, in particular neurodegenerative diseases [23]. Microglial cells are key mediators of age-related neuroinflammation, and it has been demonstrated that an increase in microglial activity can be an early event that leads to oxidative damage and cell degeneration [24]. Reactive microgliosis resulted in an increase in inflammatory cytokines [25] and increased microglial NADPH-derived ROS accumulation, which are considered to be key events in the central nervous system pathogenesis [26]. Here, we have shown that NOX4 activity is necessary for a proper inflammatory response since genetic ablation and pharmacological inhibition of NOX4 reduced NLRP3 inflammasome activation by 60% (Figure 1). Moreover, NOX4 activation is important for the expression of NLRP3 inflammasome components induced by TLR4 activation (Figure 1 and Figure 2). It has been demonstrated by different authors that there is a link between NOX4 activity and NLRP3 inflammasome activation. In Kupffer cells, NOX4-derived ROS promoted NLRP3 activation and significantly increased the expression of inflammasome components both in vitro and in vivo, aggravating liver inflammatory injury [27]. This mechanism has also been observed in various models of inflammation, such as acute pancreatitis [28], in high-glucose-induced endothelial dysfunction [29], in osteoarthritis [30], and in myocardial ischemia-reperfusion injury [31]. We also observed an in vivo anti-inflammatory effect in our model of LPS-induced memory impairment (Figure 5). In fact, genetic ablation, or pharmacological inhibition of NOX4, improved animal behavior and restored normal levels of NLRP3 inflammasome components and NOX4 (Figure 6 and Figure 7), confirming the results obtained in vitro. 

Nrf2 plays an important role in regulating oxidative stress, inflammation, mitochondrial function, and in autophagy [28]. Under normal conditions, Keap1 binds to Nrf2 in the cytoplasm and promotes its ubiquitination and proteasomal degradation [31,32]. Intracellular ROS or electrophiles alter the conformation of the Nrf2/Keap1 complex, thereby inhibiting Nrf2 ubiquitination. Thus, Nrf2 translocates to the cell nucleus, binds to the regulatory enhancer sequence ARE, and promotes the expression of antioxidant and anti-inflammatory genes [30]. On the other hand, NLRP3 also detects cellular stressors, like pathogen- and danger-associated molecular patterns (PAMPs and DAMPs) that induce inflammasome activation. The main link between Nrf2 and NLRP3 are intracellular ROS, that could come from both mitochondrial ROS and NOX activation, and can activate both proteins; however, its actions are opposite. In fact, it has been demonstrated that Nrf2 activation before NLRP3 inflammasome activation reduces inflammation in various in vitro and in vivo models of central and peripheral inflammation [33,34,35,36]. Consistent with all these studies, we have shown that 24 h treatment with tBHQ prior to NLRP3 inflammasome activation reduces NLRP3 inflammasome components both in WT and NOX4 KO mice (Figure 4). However, it is unexpected that both Nrf2 activation and Nrf2 ablation have the same consequence on the NLRP3 inflammasome activation (Figure 4). It has been recently proposed that a physical interaction of the Nrf2 complex with caspase-1 contributes to the requirement of Nrf2 expression for inflammasome activation [37,38]. Here, we showed that Nrf2 translocates to the nucleus upon LPS exposure to cells and that NOX4 activity is critical for this translocation, as Nrf2 does not translocate to the nucleus in NOX4 KO animals (Figure 3). These experiments point to NOX4 as the crucial protein for both Nrf2 translocation to the nucleus and for NLRP3 activation. Probably, other sources of ROS, like mitochondrial ROS, could be participating in this process, since NLRP3 inflammasome activation is only partially blocked in NOX4 KO mice. Hence, it is possible that the early production of NOX4-derived ROS could be a crucial step for Nrf2-NLRP3 rupture of physical interaction, but this hypothesis needs to be further investigated.

## 5. Conclusions

We present evidence for a complex crosstalk between NOX4, Nrf2, and NLRP3 inflammasome pathways. We contribute to the understanding of the role of NOX4 and Nrf2 in the activation of the NLRP3 inflammasome. We demonstrate that NOX4 is a key player in NLRP3 inflammasome activation suggesting NOX4 pharmacological inhibition as a potent therapeutic approach in neurodegenerative diseases. 

## Figures and Tables

**Figure 1 antioxidants-12-01729-f001:**
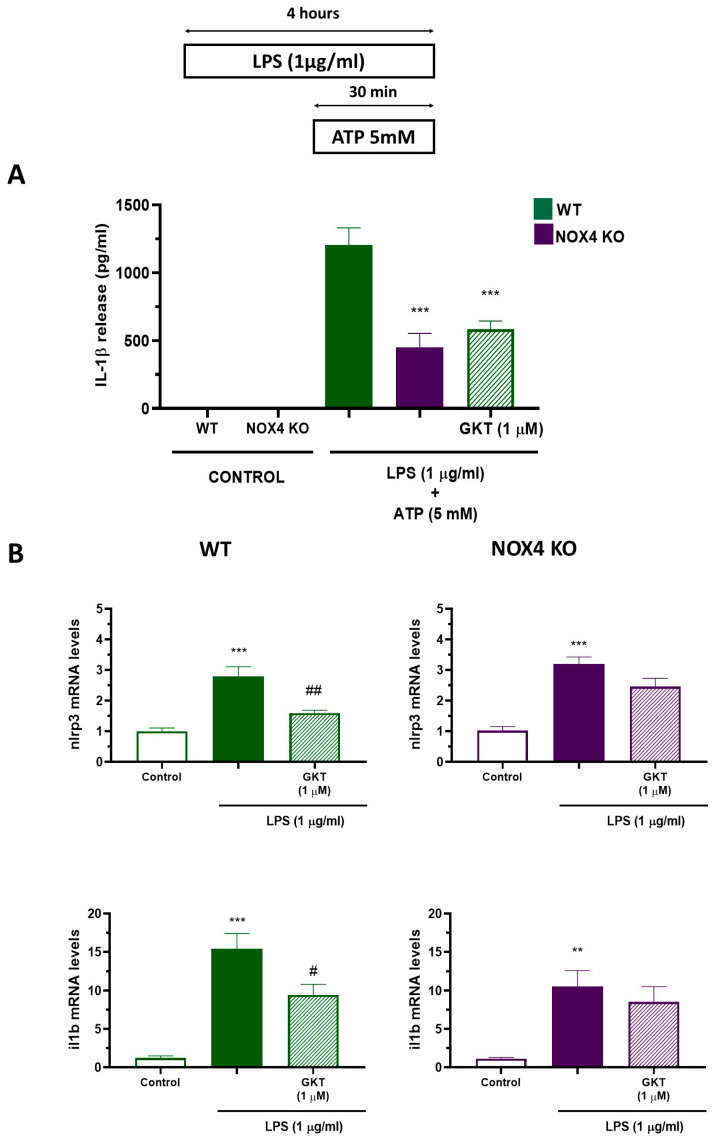
Nox4 activity is required for NLRP3 inflammasome activation. (**A**) Mixed glial cultures of WT and NOX4 KO mice were stimulated with LPS (1 µg/mL) for 4 h plus ATP (5 mM) during the last 30 min, in the absence or presence of GKT136901 (1 µM) following the protocol at the top of the figure. Inflammasome activation was analyzed via ELISA measurement of IL-1β in the supernatant. Mean ± SEM (*n* = 6). Two-way ANOVA followed by Bonferroni’s tests *** *p* < 0.001 vs. WT LPS. (**B**) At the end of the experiment, cells were harvested and analyzed for the expression of NLRP3 inflammasome components Nlrp3 and Il1b via RT-PCR. Mean ± SEM (*n* = 5). One-way ANOVA with Tukey’s multiple comparisons test. *** *p* < 0.001 and ** *p* < 0.01 vs. control; ## *p* < 0.01 and # *p* < 0.05 vs. LPS-treated cells.

**Figure 2 antioxidants-12-01729-f002:**
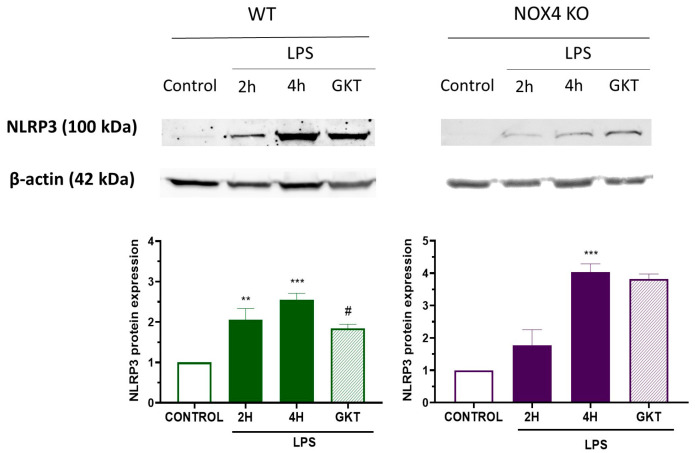
NLRP3 levels in mixed glial cultures of WT and Nox4 KO mice after LPS treatment. Mixed glial cultures of WT and NOX4 KO mice were stimulated with LPS (1 µg/mL) for 2 and 4 h, in the absence or presence of GKT136901 (1 µM) following the protocol at the top of Figure 1. Changes in NLRP3 protein amounts were determined via Western blot. A representative Western blot image of NLRP3 protein and actin is shown (top). Mean ± SEM (*n* = 5). One-way ANOVA with Tukey’s multiple comparisons test. *** *p* < 0.001 and ** *p* < 0.01 vs. control; # *p* < 0.05 vs. 4 h LPS-treated cells.

**Figure 3 antioxidants-12-01729-f003:**
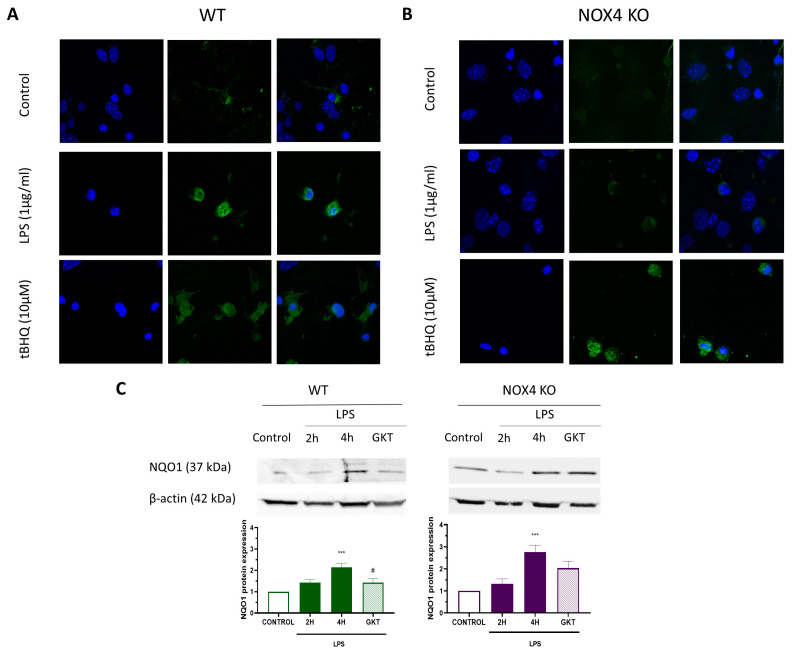
LPS treatment for 4 h induces Nrf2 translocation to the nucleus and NQO1 protein levels. High magnification confocal images of mixed glial cultures from WT (**A**) and NOX4 KO (**B**) immunostained with Nrf2 (green) and counterstained with DAPI (blue) to illustrate nuclei. Mixed glial cultures were untreated (Control), treated with LPS (1 µg/mL) for 4 h or treated with tBHQ (10 µM) for 24 h. (**C**) NQO1 protein levels in WT and NOX4 KO mice. Animals were treated with LPS (1 µg/mL) for 2 and 4 h, in the absence or presence of GKT136901 (10 µM). A representative Western blot image of NQO1 is shown at the top. Mean ± SEM (*n* = 5). One-way ANOVA with Tukey’s multiple comparisons test. *** *p* < 0.001 vs. control; # *p* < 0.05 vs. LPS-treated animals.

**Figure 4 antioxidants-12-01729-f004:**
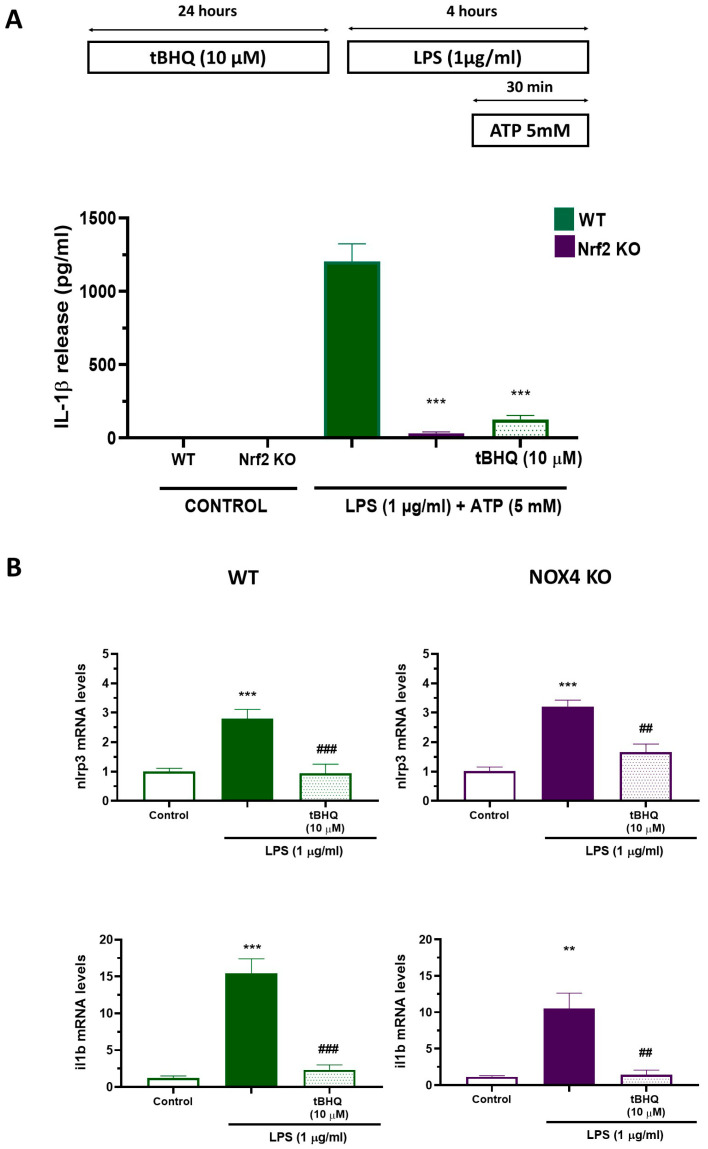
Nrf2 activity is required for NLRP3 inflammasome activation. (**A**) Mixed glial cultures of WT and Nrf2 KO mice were treated with tBHQ (10 µM) for 24 h followed by stimulation LPS (1 µg/mL) for 4 h plus ATP (5 mM) during the last 30 min, following the protocol at the top of the figure. ELISA measurements were performed for quantification of IL-1β secretion. Mean ± SEM (*n* = 5). Two-way ANOVA followed by Bonferroni’s tests *** *p* < 0.001 vs. WT LPS. (**B**) Mixed glial cultures of WT and NOX4 KO mice were treated with tBHQ (10 µM) for 24 h followed by the stimulation with LPS (1 µg/mL) for 4 h. At the end of the experiment, cells were harvested and analyzed for the expression of NLRP3 inflammasome components Nlrp3 and Il1b by RT-PCR. Mean ± SEM (*n* = 5). One-way ANOVA with Tukey’s multiple comparisons test. *** *p* < 0.001 and ** *p* < 0.01 vs. control; ### *p* < 0.001 and ## *p* < 0.01 vs. LPS-treated cells.

**Figure 5 antioxidants-12-01729-f005:**
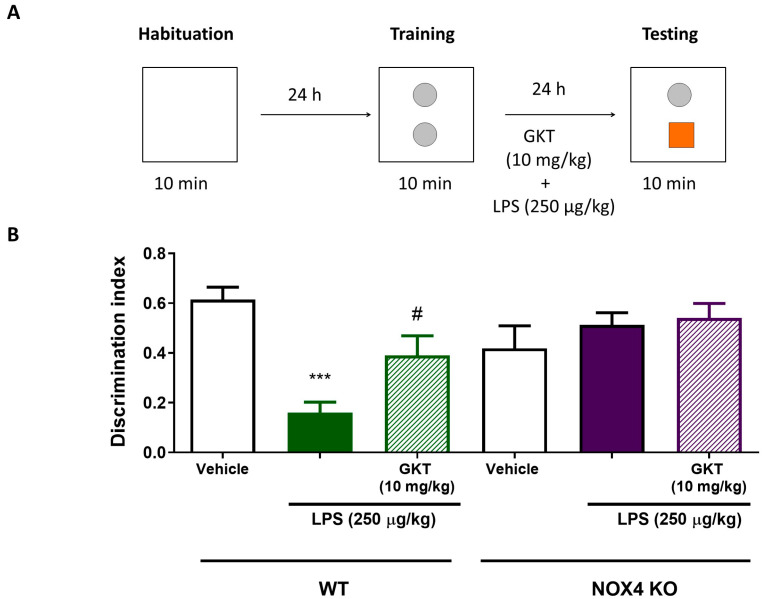
Genetic ablation and pharmacological inhibition of Nox4 improves LPS-induced memory impairment. (**A**) Illustration of the protocol used to evaluate novel object recognition (NOR) test. WT and Nox4 KO mice were treated with LPS (250 μg/kg, i.p.) alone or in combination with GKT (10 mg/kg, i.p.) and after 24 h were subjected to the test. (**B**) Discrimination index in WT and Nox4 KO mice in saline-treated, LPS-treated and GKT-treated groups. Mean ± SEM (*n* = 7). Two-way ANOVA followed by Bonferroni’s tests *** *p* < 0.001 vs. WT vehicle and # *p* < 0.05 vs. WT LPS.

**Figure 6 antioxidants-12-01729-f006:**
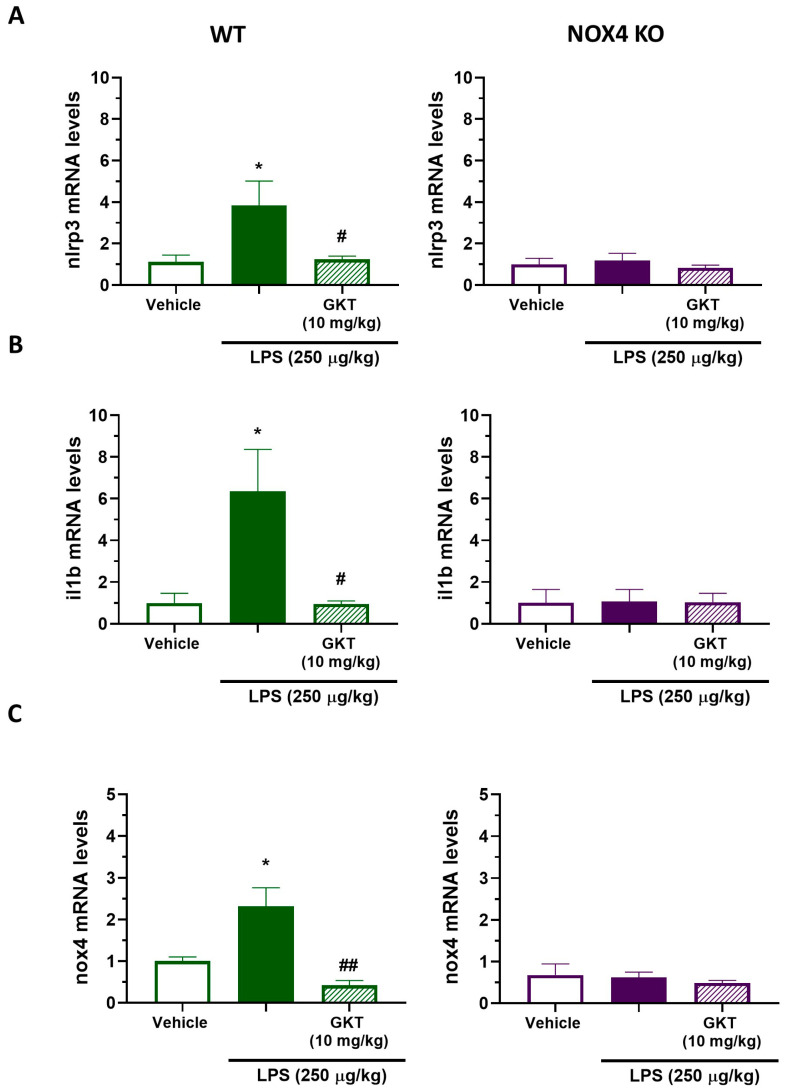
NOX4 activity is necessary for NLRP3 inflammasome components and Nox4 expression in hippocampus of LPS-treated mice. Expression of genes of the inflammasome NLRP3 complex and NOX4 in the hippocampus of WT and Nox4 KO mice after 24 h LPS and/or GKT treatment. mRNA levels of inflammasome components Nlrp3 (**A**), Il1b (**B**), and Nox4 (**C**) in the hippocampus of WT (*n* = 5) and Nox4 KO (*n* = 5) mice treated with LPS (250 µg/kg) and/or GKT (10 mg/kg) for 24 h. Mean ± SEM (*n* = 5). One-way ANOVA with Tukey’s multiple comparisons test. * *p* < 0.05 vs. vehicle; ## *p* < 0.01 and # *p* < 0.05 vs. LPS-treated animals.

**Figure 7 antioxidants-12-01729-f007:**
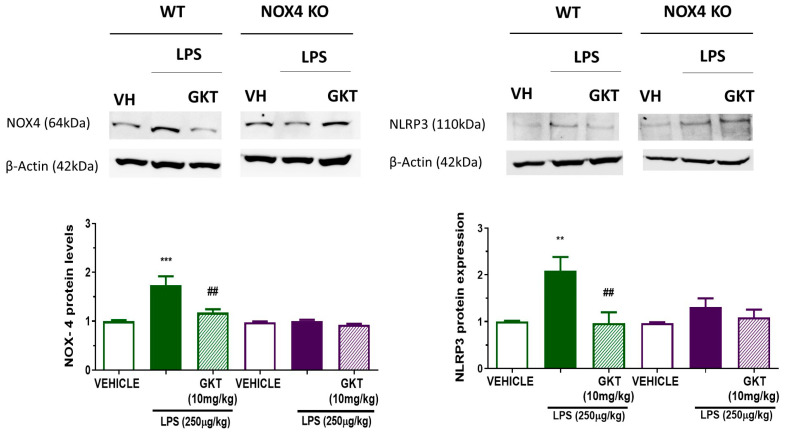
NOX4 and NLRP3 protein levels in the hippocampus of WT and NOX4 KO mice treated with LPS for 24 h. Changes in Nox4 (**left**) and NLRP3 (**right**) levels in WT and NOX4 KO mice. Animals were treated with LPS (250 µg/kg) for 24 h, in the absence or presence of GKT136901 (10 mg/kg). Representative Western blot image of Nox4, NLRP3, and actin is shown on top. Mean ± SEM (*n* = 5). One-way ANOVA with Tukey’s multiple comparisons test. *** *p* < 0.001 and ** *p* < 0.01 vs. control; ## *p* < 0.01 vs. LPS-treated animals.

## Data Availability

The data presented in this study are available on request from the corresponding author.

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
