# Peer review of "Redox Regulation of Microglial Inflammatory Response: Fine Control of NLRP3 Inflammasome through Nrf2 and NOX4"

_antioxidants, 2023, doi:10.3390/antiox12091729_

Round 1

Reviewer 1 Report

The authors We hypothesized that the interplay between Nrf2 and NOX4 may play a critical role in the activation of the NLRP3 inflammasome and subsequent inflammatory responses. Mixed glial cultures of NOX4 and Nrf2 KO mice were stimulated and inhibited by various drugs showing significant changes of IL1-β liberation as well as in vivo mouse behavioural experiments were evaluated in order to proof the hypothesis.

The experiments are interesting and performed sound using various methods. All in all the paper is of good quality.

Nevertheless, there are some critical points to be mentioned and – consequently - to be improved.

Abstract:

-please do not used abbreviations in the abstract

-You say: mice treated with GKT136901 after LPS impairment showed significantly improved discrimination index and recover the expression of inflammatory genes to normal levels. Question: Compared with what?

Introduction:

-Explain Nrf2

-otherwise, clear

Materials and Methods:

-give date and number of permission of experiments

-2.6. …what about the hippocampus – not mentioned before – preparation must be described in M+M

Results:

-Fig 1: x- and y-axis nearly unreadable – pictures are too small – this is true for nearly all graphs (Fig 5 is ok)

- Font size and font must be consistent in all graphs

-You say: Moreover, we observed the same effect using GKT136901 (1 μM), a selective pharmacological inhibitor of NOX4 (Figure 1A). Question: do you mean WT-mice cultures?

-Fig2: the ß-actin lane in WT GKT is not really typical for the result shown in the respective bar chart

-Fig7, left: lanes in WT and KO are not really typical for the result shown in the respective bar charts

-otherwise, clear

Discussion:

appropriate and understandable

Reviewer 2 Report

This is in principle an interesting and well-conducted study uncovering novel associations between NOX4, NRF2 and the NLRP3 inflammasome which however requires some revision.

1.       Line 61: Please explain what a “pyroptotic cell” is and how NLRP3 activation induces it.

2.       Line 67: importantly, also cellular senescence is induced by ROS in addition to apoptosis (by very similar pathways)

3.       Line 72: I think it is incorrect that NOXes generate the majority of cellular ROS since more than 90% are generated by mitochondria. Please correct.

4.       Line 91: Since NRf2 activates antioxidant enzymes, it does NOT reduce ROS production which is completely independent from any antiox enzyme. They just change ROS LEVELS!

5.       Line 134: Please describe what your vehicle is (PBS?) and how you applied the agents (i.p.)

6.       Line 143: Please describe what the novel object was and also add it into the figure.

7.       Line 192: Please describe what microscope (was it also a Leics SP5 confocal as described for IHC in line 208?), magnification etc were used and how many field of vision were analysed per sample.

8.       Figure 1: Please do not show the figure under “methods” but under “results”. In A: Please label what the line under “control” are: wt and ko? And also make clear that the inhibitor was used in wt by including it into the colour labels. In B (like in many other graphs): axis labels are too small. Please also describe what your n-number refers to: are these 6 independent cultures derived from 6 different mice or are these just technical repeats of the same culture?

9.       Line 251, 254, 281, 288, 346 and throughout the manuscript: Please do not use the term “expression” for proteins since ONLY genes are expressed. You confuse the reader since often it is not clear whether you refer to RNA (PCR) or proteins. Use “level/amounts” for proteins and RNA and “expression” exclusively for genes.

10.   Figure 2: Make axis labels larger in the graphs.

11.   Line 266: Please add an appropriate reference that evidences the predominant importance of NOX4.

12.   Line 268: Please explain what a “phase II response “is.

13.   Line 283: ROS are not “activated” but generated. There is no activation process. Please rephrase and use “generation”!

14.   Line 283: Why does NRF2 which activates antiox enzymes, “participates” in NOX4-derived ROS generation? This does not seem to make sense as both factors (NRF2 and NOX4 seem to counteract each other. For somebody who is not directly from this field, this does not seem logical. Please give more details of this rational as well as of the function of NQO1.

15.   Fig. 3C: make axis labels larger as they are hardly readable. Also, again, please explain what exactly your n-numbers represent.

16.   Line 297: Please mention that with “the same blocking” you refer to both genotypes.

17.   Figure 4A: the scheme in invisible, font needs to be larger.

18.   Lines 299-304: while you initially talk about NLRP3 inhibition you then state about its activation without giving any further explanation. For many readers this might seem confusing.

19.   Fig. 4: In A, same as in fig. 1A: show what genotypes the controls are instead of giving just empty labels. 4B: enlarge axis labels. Also, explain what your n- numbers are: cultures from different mice or technical repeats from the same cultures?

20.   Throughout the manuscript: since you always use the same concentrations of LPS and inhibitors, there is no need to give these over and over again in the text and figure legends. It is in the methods and this is sufficient. It just disrupts the flow of reading.

21.   Statement in line 322 requires a reference.

22.   In figure 5 A: show the novel object for the testing stage and not just the old object and an empty space!

23.   Line 346/7: since your measured expression by PCR it is “gene expression”, NOT expression of proteins!

24.   Figure 6: enlarge axis label font.

25.   Line 359: add “WT

26.   “ in front of “animals”

27.   Line 362: Since you analyse proteins here by WB, do not use “expression”, but “protein levels”

28.   Line 365/6: do not use “expression”, but “protein levels”

29.   Figure 7: enlarge axis font for the graphs and also use “protein levels” for the right graph (NLRP3).

30.   Line 380: Please better explain where the TLR4 activation comes from as you haven’t mentioned this receptor previously.

31.   Line 384: “index IN the NOR test” to make it clearer what the index refers to.

Language mistakes:

Line 32: replace “liberation” with “RELEASE”. Line 38: add “mouse” in front of “hippocampus” and remove “the”. Line 39: add “the” between to and “control. Line 56: it should be “interleukin”. Line 61 “and pyroptotic cell” is incorrect grammar. Either “a… cell” or “… cellS”. Line 75 “ to produce A superoxide anion”. Line 94: “ii) IN …animals…”. Line 101: “compared to THE control group”. Line 102: “showed A significantly…” and “recovery”. Line 242: The idiom “deepen the mechanism” is not correct grammar/language; better say “to better explore the mechanism”. Line 261: “NOX4 activity RESULTED IN NRF2 translocation to the nucleus THAT is necessary….”-your grammar is wrong. Line 263: “As stated in THE introduction…”. Line 267: “regulator of antioxidant (remove article) and… responses”. Line 268: also remove “the” here in front of “phase II”. Line 275 “but NOT in…”, line 277: “tBHQ TREATMENT RESULTED IN”. line 280: “by immunoblot” needs to be at the end of the sentence as the explanation refers to NQ01, not the blot. Line 281: “increased NQO1 protein levels by 2 fold” (word order and replace “expression” for proteins”. Line 283:”participates IN…”. line 298: again, replace “deepen” with “explore further”. Line 300: replace “produced” with “resulted in AN increase of…”. Line 315: Grammar of the heading is wrong: “… RESULTS in memory….”. line 339: it should be “expression of components” or “component expression”. line 344: “MOUSE hippocampus”Line 345: “there WAS a…increase”. line 350: “componentS”. Line 357: “in THE hippocampus”. Line 374: “understanding OF the role…”. Line 383: “compared to THE control”. Line 384: “showed A…” “…and recovered” or “RECOVERY OF”. Line 420: “CONSISTENT with” -no adverb required here (used only together with a verb). Line 421: please replace “works” with STUDIES. Line 432: order of words “OS ONLY”

English is in general ok, just a few minor grammar errors (lack of articles or other words) is required that I mention to the authors in my report.

Reviewer 3 Report

In this study, Palomino-Antolí et al. investigate the role of NOX4 and Nrf2 during Inflammasome activation in microglia. The introduction and the discussion are excellently written and the topic of the study is very interesting, so I was very looking forward to read this article. However, the massive flaws in the experimental design and obvious contraries in the results leave me no other option as to reject this manuscript at this point in time.

Round 2

Reviewer 1 Report

Dear authors,

thank you for improving the manuscript according to my comments and suggestions.

You did a good job.

Author Response

Thank you for your useful comments

Reviewer 3 Report

I thank the authors for clearling things up. These were important methodical informations. It is very interesting that the Nox4 was only modified in terms of its activity. This article makes much more sense now. However, in this new light ROS measurements (with 6-Carboxy-DCF) in vitro with glial cells (WT and Nox4-KO) are a crucial control to prove that the enzyme is present, but not working as the authors claim. You can use PMA as general positive control. Please perform a kinetic (60 mins). If the authors can provide these experiments, this article we be a helpful contribution to the field.

Round 3

Reviewer 3 Report

There was a litte bit of confusion for the interpretation of my suggestion.

Of course, PMA is only a positive control for the DCF measurement itself and activates, like the authors rightly claimed, many NOX isoforms.

The experiments with WT and NOX4 cells should be performed with the stimuli used in the experiments, in this case LPS+ATP and with the NOX4-Inhibitor.

Author Response

We sincerely appreciate your insightful feedback and the opportunity to address your concerns involving WT and NOX4 cells treated with LPS+ATP stimulation and a NOX4-inhibitor. We understand the importance of rigorous experimental design and are pleased to provide a comprehensive justification for our decision not to include a control experiment in this specific context. Given the robustness and consistency of these findings, the inclusion of the suggested experiments would not provide novel insights or contribute substantially to the existing body of knowledge. In any case, we have included a brief paragraph with a special emphasis on the limitations of the experiment and an adequate exposition of the selection of the experimental design used into the methods section and into the discussion.

Thank you once again for your valuable feedback and consideration.